# Mind the Budget: Accelerating Deep Reinforcement Learning using Constrained Early Exit Neural Networks

**Julien Broßeit** [1]   **Jasper Hoffmann** [1]   **Joschka Boedecker** [1]

## Abstract

Early exit neural networks, which adapt computation to input complexity, have proven effective in supervised learning but remain largely unexplored in deep reinforcement learning (DRL). In this paper, we propose Budgeted EXit Actor (`BEXA`), a novel actor-critic architecture that integrates early exit branches into the actor network. To ensure a mathematically principled trade-off between performance and inference expenditure, the exit decision is formulated as a constrained linear program during training, the solution of which is subsequently amortized to enable highly efficient runtime execution. `BEXA` is general, governed by an interpretable budget parameter, and compatible with any off-policy actor-critic method. We evaluate `BEXA` by integrating it with standard off-policy algorithms like SAC and TD3 on a suite of MuJoCo tasks. Our results demonstrate a substantial improvement in inference efficiency with minimal or no loss in performance. These findings highlight early exits as a promising direction for improving computational efficiency in DRL.

## 1. Introduction

Recent work has demonstrated favorable scaling properties of large neural networks (NNs) in deep reinforcement learning (DRL), (Farebrother et al., 2024; Nauman et al., 2024; Obando-Ceron et al., 2024). However, increasing network depth leads to higher computational costs, making DRL more expensive to train and more difficult to deploy. This is particularly problematic in robotics, where strict constraints on inference time must be met, or when deploying fine-tuned LLMs that demand immense computational resources. Current methods in DRL try to speed up inference by using

[1]Neurorobotics Lab, University of Freiburg, Germany. Correspondence to: Julien Broßeit <brosseit@informatik.uni-freiburg.de>.

*Proceedings of the 43rd International Conference on Machine Learning*, Seoul, South Korea. PMLR 306, 2026. Copyright 2026 by the author(s).

model compression techniques like neural network pruning and quantization, reducing the number of model computations while maintaining performance (Zhang et al., 2023). Nevertheless, these approaches are sensitive to tuning and can introduce training overhead, particularly in the context of quantization (Fu et al., 2025).

Importantly, a lot of compression methods fail to leverage an inherent property of function approximation in DRL: the computational complexity required for selecting an optimal action varies with the state. For illustration, in chess, finding the best move depends on the complexity of the position. Some positions allow for quick detection of strong moves, while others require extensive computation. Thus, in DRL, where the policy is a neural network, we face a challenge: traditional neural networks are static, performing the same computations regardless of the input. This can lead to inefficiency, since for some inputs an action could be derived with significantly fewer computations.

Early exit neural networks (ENNs) are dynamic NNs that adapt their computational graph based on the input. Originating in fields with high computational demand like computer vision (CV) (Laskaridis et al., 2021) and natural language processing (NLP) (Xu & McAuley, 2023), they work by adding side branches to the network, so-called exits. A gating mechanism decides on which exit to take, adaptively controlling the network depth, enabling a trade-off between performance and efficiency, and potentially improving generalization and interpretability (Han et al., 2022). In many CV and NLP tasks, they have achieved performance comparable to that of their static counterparts while using only a fraction of floating point operations (FLOPs).

Despite their advantages, dynamic neural networks like ENNs have been hardly explored in reinforcement learning (RL). A partial explanation is that naively applying such networks to RL is not possible, as RL has some unique aspects compared to supervised learning. One of the biggest challenges is the lack of supervision, as the agent has to find the correct actions on its own. Typically, early exit branches are trained directly on fixed ground truth data, whereas in RL the behavior of the agent changes over time. In addition, the predicted actions influence the state distribution encountered by the agent. Poorly chosen actions of ENNs can therefore

lead to learning instabilities.

In this work, we systematically investigate how to transfer ENNs into DRL. Based on our findings, we propose a new method called Budgeted EXit Actor (`BEXA`), which introduces early exit NNs with resource-constrained gating based on Q-values to speed up policy inference time during training and evaluation. Our main contributions are:

1. We present `BEXA`, a general off-policy actor-critic method, with careful adjustments for using early exit neural networks (ENNs) effectively in DRL for reducing the number of required FLOPs during training and deployment.

2. By formulating sequential exit decisions as a constrained linear program during training and subsequently amortizing the solution, we introduce a principled framework that optimizes the trade-off between expected return and computational costs without online optimization overhead.

3. We integrate `BEXA` into standard continuous control algorithms (SAC and TD3) across a suite of MuJoCo tasks. Our results demonstrate substantial inference and sample collection speedups with minimal to no loss in return, while systematic ablations validate that our joint training objective and optimization-driven gating outperform alternative heuristics.

## 2. Related Work

We divide related work into two categories: (i) early exit neural networks (ENNs), which have been primarily explored in domains outside of deep reinforcement learning (DRL), and (ii) methods for accelerating training and inference in DRL, such as model compression and software optimization. Within the second category, we also highlight the few approaches that combine both directions in a manner similar to our work.

### 2.1. Early Exit Neural Networks

Early exit neural networks (ENNs) belong to the family of dynamic neural networks (NNs). These are models that change their computational graph based on the input they receive (Han et al., 2021). Some instances adjust their depth using early exits, while others adjust their width by changing the number of neurons or channels in each layer, or by changing their parameters. We focus on early exit networks because they are conceptually straightforward and have been extensively studied (Scardapane et al., 2020b). Here, we will present only a few noteworthy works and refer the interested reader to comprehensive surveys (Laskaridis et al., 2021; Xu & McAuley, 2023; P et al., 2025).

The first works for these networks include conditional deep learning network (CDLN) (Panda et al., 2016) and BranchyNet (Teerapittayanon et al., 2016). CDLN first trains the backbone network and then adds linear early exits at multiple depths, retaining only those that improve performance. BranchyNet integrates exits into known computer vision (CV) classifier networks and uses an entropy-based criterion to terminate computation early. All exits are trained jointly with a weighted cross-entropy loss. While effective in CV, such entropy criteria are not directly applicable to DRL, where high policy entropy is beneficial for exploration.

More recent work goes beyond simple entropy-based criteria with alternative decision rules. Confidence to exit can be defined by maximum class probability (Huang et al., 2018; Wang et al., 2022) or by patience, exiting when several consecutive branches agree (Zhou et al., 2020; Zhu, 2021), a strategy common in early exit transformers. Beyond heuristics, the decision to exit can also be learned: Demir & Akbas (2024) jointly optimizes accuracy and efficiency to train exits and gates, while Vashist et al. (2022) uses DRL to learn an exit policy using a deep Q-network (DQN), though the underlying task is not a DRL one. The work presented here, can be seen as an extension to tuning the exit selection process with DRL. However, rather than learning gate decisions with reinforcement learning (RL), we formulate exit selection as a resource allocation problem: value estimates from DRL are optimized under a budget constraint via a linear program, which supervises gate policy learning.

Training strategies can also vary. The most common approach is to jointly optimize all exits under a combined loss (Berestizshevsky & Even, 2019; Scardapane et al., 2020a). However, the modular design of early exits also allows for a layer-wise training scheme (Hettinger et al., 2017), where a subset of exits is trained at a time while keeping the rest frozen.

Finally, self-distillation (Zhang et al., 2019) is a variant of knowledge distillation in which knowledge is transferred from a teacher model to one or more student models. Applied to an ENN, the final output layer can be considered the teacher, while the intermediate outputs are the students. These outputs are trained using a combination of a standard supervised loss and an additional imitation loss that encourages the outputs to mimic the teacher's predictions. Previous work has shown that self-distillation can improve model accuracy (Pham et al., 2022) also in the context of ENN (Zhang et al., 2022).

### 2.2. Accelerating DRL

Two directions have emerged for accelerating training and inference in DRL. The first targets system-level efficiency through software and hardware optimizations, such as paral-

lelization and the use of accelerators like GPUs. The second approach focuses on model-level efficiency, compressing neural networks that represent policies, value functions and dynamics models.

On the system side, Weng et al. (2022) parallelizes environment simulation with a C++ backend, reducing Python overhead and enabling high-throughput sampling. We adopt this setup in our experiments as well. Pushing this further, Dalton & Frosio (2020) ports Atari to the GPU, yielding even faster parallel roll-outs. Architecturally, IMPALA (Espeholt et al., 2018) decouples acting from learning by running environments in separate processes, each with its own policy, and asynchronously aggregates experience into a shared buffer. SEED RL (Espeholt et al., 2019) refines this design by batching observations from many environments and evaluating a single policy on an accelerator throughout training.

On the model side, NN compression techniques such as quantization (Nagel et al., 2021), knowledge distillation (Hinton et al., 2015), and pruning (LeCun et al., 1989) have proven highly effective in supervised learning for improving runtime. Recent works adapt these techniques to DRL. QuaRL (Krishnan et al., 2022) quantizes policy parameters after each update from 32-bit floating point to 8-bit integers, improving throughput with minimal accuracy loss. FastAct (Zhang et al., 2023) generalizes this idea by supporting arbitrary compression schemes, while a scheduler ensures that compression remains within acceptable limits to maintain performance.

One closely related line of work is RAPID-RL (Kosta et al., 2022), which integrates early exit networks into DQN. It estimates confidence by checking whether an exit's Q-value exceeds a fixed fraction of the maximum Q-value, employs layer-wise training, and reports faster inference on Atari. Our approach differs in three key aspects: (i) we target general actor–critic methods rather than DQN, requiring early exits only for the actor and permitting more flexible critic architectures, (ii) we introduce a novel resource-aware early exit criterion and train it jointly with all exits, and (iii) whereas RAPID-RL primarily reduces deployment-time inference but incurs training overhead by evaluating all exits, our method accelerates both training and inference.

## 3. Preliminaries

Reinforcement learning (RL) problems are commonly formulated as Markov decision processs (MDPs). An MDP consists of an agent interacting with an environment, where the agent follows a policy $\pi(a \mid s)$ that determines the next action given a state $s$. At each time step $t$, the agent chooses an action $a_t$ that is executed in the environment, which, in response, returns the next state $s_{t+1} \sim P(s_{t+1} \mid s_t, a_t)$

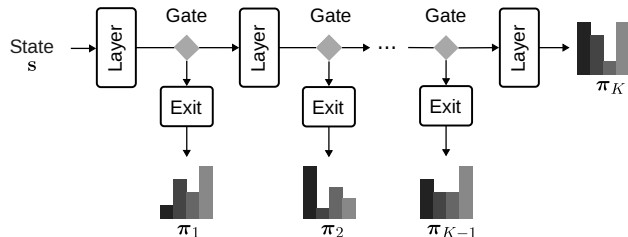

Figure 1. Example of an actor represented as an early exit neural network (ENN). The input state is processed layer-by-layer until a gate is reached. Based on a learned rule, a gate decides whether to terminate early or proceed with the computation. Each exit produces an action distribution for the current input state.

according to the transition probability function $P$. Additionally, the agent receives a reward $r_t = R(s_t, a_t) \in \mathbb{R}$, where $R$ is the reward function. This reward is a scalar value that describes the desirability of the given state and the chosen action. The cumulative sum of rewards, known as the return, is defined as $G_t = \sum_{k=0}^{\infty} \gamma^k r_{t+k}$ where $\gamma \in [0, 1]$ is the discount factor that determines the importance of future rewards. We define the state-value function $V_\pi(s) = \mathbb{E}_\pi[G_t \mid s_t = s]$ to calculate the expected return following some policy $\pi$. Similarly, we define the action-value function $Q_\pi(s, a) = \mathbb{E}_\pi[G_t \mid s_t = s, a_t = a]$ as the expected return if first an initial action $a$ is taken, after which the policy $\pi$ is followed. Given an initial state distribution $\rho_0$, the goal in RL is to find an optimal policy $\pi^*$ that maximizes the expected return $\pi^* \in \arg\max_\pi \mathbb{E}_{s \sim \rho_0}[V_\pi(s)]$.

In deep reinforcement learning (DRL), policies and value functions are typically represented by a deep neural network (NN): the policy (actor) $\pi^\theta$ with parameters $\theta$, and the action-value function (critic) $Q^\phi$ with parameters $\phi$. Actor-critic methods jointly learn both networks, where the policy typically maximizes an objective derived from the critic that is of the form $J_\pi(\theta; Q^\phi) = \mathbb{E}_{s \sim \mathcal{D}}\big[\mathbb{E}_{a \sim \pi_\theta(\cdot|s)}[Q^\phi(s, a)]\big]$, where $\mathcal{D}$ is a replay buffer collecting states from interactions with the environment. Many state-of-the-art algorithms, such as SAC (Haarnoja et al., 2018) and TD3 (Fujimoto et al., 2018), are off-policy actor-critic methods, meaning they learn from data $\mathcal{D}$ collected by past policies rather than requiring samples from the current policy.

## 4. Method

We now present our framework for integrating an early exit neural network (ENN) into off-policy actor-critic methods. The approach is general and can be applied to any actor-critic method with minimal changes to the underlying architecture. We first introduce the early-exit actor architecture, then describe how exit selection is formulated as a budget-constrained resource allocation problem, and finally

present the complete algorithm.

### 4.1. Early Exit Actor

The key distinction in our approach is that we represent the actor as a deep ENN shown in Fig. 1. During the forward pass, data is propagated sequentially through the network layers. At each side branch, a gating policy decides whether to terminate the computation early. A stochastic gating rule is used instead of a deterministic one to avoid premature collapse to a small subset of exits and ensure that each exit is occasionally selected. If the gate activates, the corresponding early exit head is evaluated and its prediction is returned without subsequent layers being evaluated, thereby saving computation. Otherwise, the computation continues and the exit is not calculated. To reduce computational overhead, the gating function shares its hidden features with the actor.

We now formalize the architecture mathematically. First, we number the exits sequentially from earliest to last and denote the sub-policy at each exit by $\pi_i$ for $i = 1, \ldots, K$. Additionally, each exit before the final layer has a gate policy $g_i(\cdot \mid s) = \text{Bernoulli}(p_i(s))$, where $p_i$ is a learned state-dependent probability parameter; sampling 1 indicates taking the exit, while 0 means resuming. Given a state $s$, the probability $\alpha$ that we terminate at exit $i = 1, \ldots, K$ is given by

$$\alpha_i(s) := p_i(s) \prod_{j<i} \big(1 - p_j(s)\big), \tag{1}$$

assuming that we define $p_K(s) = 1$. The resulting policy $\pi$ represented by the entire ENN is then a mixture of the exit policies

$$\pi(a \mid s) := \sum_{i=1}^{K} \alpha_i(s)\, \pi_i(a \mid s).$$

### 4.2. Learning Budget-Aware Early Exit Actors

We now discuss how to learn the gating policies $g_i$. In the supervised learning setting, ENNs typically rely on confidence-based exit rules such as measuring the entropy of the prediction, maximum class probability or patience, a criterion that exits once at least $n$ consecutive predictions align (Xu & McAuley, 2023). These criteria are ill-suited for deep reinforcement learning (DRL), as high entropy drives exploration, which is crucial for success. Applying confidence-based methods steers behavior toward greedy action selection. Patience is also ineffective because DRL models are usually smaller than those in natural language processing (NLP) and offer much fewer exits. Such methods also introduce task-specific hyperparameters like thresholds that are difficult to tune.

In our reinforcement learning (RL) setting, each exit defines a policy $\pi_i$, and we can compare their performance directly

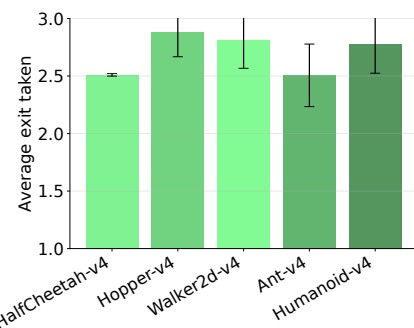

*Figure 2.* Average selected exit during soft actor-critic (SAC) training with greedy selection of exits based on learned Q-values. Results show the mean of the top five out of 200 agents per environment with one standard deviation. This early exit network has three branches $K = 3$. Without a resource constraint, later exits dominate.

using the expected value $V_{\pi_i}$. A natural idea is to pick the exit that maximizes the expected value in the current state, i.e., $\arg\max_i V_{\pi_i}(s)$. However, this approach tends to favor later exits, as they build upon the representations of previous layers and generally achieve higher returns. This intuition is supported by an experiment shown in Fig. 2, where later exits are selected disproportionately often, resulting in only minimal speedups. Moreover, this method provides little explicit control over the trade-off between performance and computational cost. Thus, we need to explicitly constrain the usage of later exits.

**Constrained Optimal Budget-Aware Exit Selection.** Different from approaches that rely on heuristics that are potentially hard to tune, we propose a principled approach that formulates early exit selection as a resource allocation problem. The key idea is that we maximize the expected value of the network's actions while enforcing a hard budget constraint on inference costs.

Let $Q_i$ be the Q-function of each exit policy $\pi_i$. Given a state $s$, let $\mathbf{v} = [Q_1(s, a_1), \ldots, Q_K(s, a_K)]^\top$ with $a_i \sim \pi_i(\cdot \mid s)$ be an unbiased value estimate for each exit, and let $\mathbf{c} = [c_1, \ldots, c_K]^\top$ denote the per-exit costs, e.g., their floating point operations (FLOPs). For a given budget $b \in \mathbb{R}$, we then solve the following linear program:

$$\alpha^\star = \arg\max_{\alpha \in \mathbb{R}^K} \quad \mathbf{v}^\top \alpha \tag{2a}$$

$$\text{s.t.} \qquad \mathbf{c}^\top \alpha \le b, \quad \alpha \ge 0, \quad \mathbf{1}^\top \alpha = 1 \tag{2b}$$

The optimal weighting vector $\alpha^\star$ denotes the optimal probability distribution over exit choices that maximizes the expected value while keeping the total cost within the budget. By rearranging Eq. 1 we obtain the optimal

probabilities $p_i^\star$ for each gate. The cost definition corresponds to the resource of interest, for example FLOPs with $c_i \propto \text{FLOPs}(\pi_i)$ though other choices are also possible. For our approach we decided to use normalized FLOP counts with $c_1 = 0$ and $c_K = 1$ and scale the intermediate costs linearly, while still satisfying $c_1 \le c_2 \le \cdots \le c_K$. The scalar budget $b$ specifies a limit on the usable resources. With normalized costs $b \in [0, 1]$ becomes intuitive to scale. As $b$ approaches zero, the gate favors earlier exits, while for $b$ approaching one, the gate prefers the later exits. This yields a direct and tunable trade-off between speed and performance.

Lastly, we note two points. First, the formulation and its solution $a^*$ are state-dependent, ensuring that the expected compute satisfies the budget constraint for each state. Second, the linear program in Eq. 2 is efficient to solve, since $K$ is small in practice and the candidate extreme points can be enumerated quickly.

**The BEXA Training Objective.** Finally, we present the complete learning framework, which uses the optimal gate probabilities $p_i^\star$ as a supervisory signal. To keep our approach general, we assume that for a parameterized policy $\pi^\theta$ and critic $Q^\phi$ the underlying actor-critic algorithm provides an actor objective $J_{\text{actor}}(\theta; \pi^\theta, Q^\phi)$ that should be maximized with respect to the parameters $\theta$.

For each exit policy $\pi_i^\theta$, we learn a corresponding critic $Q_{\pi_i}^\phi$. Preliminary experiments indicated that maintaining a separate critic per exit substantially improved learning stability. Since the underlying method is off-policy, each critic can be learned from the same stream of data. To avoid the computational cost of $K$ separate critics, we use a single critic with shared features and $K$ heads, one per exit, which adds only minor overhead. Importantly, we do not impose an early exit structure on the critic, which can lead to significant instability during training.

The final Budgeted EXit Actor (BEXA) method trains the complete early exit actor by optimizing an actor-critic objective $J_{\text{actor}}$ and a gate loss $\mathcal{L}_{\text{gate}}$ at every exit. Thus, combined, we maximize the following objective for the actor:

$$J_{\text{BEXA}}(\theta) = \sum_{i=1}^{K} \left( J_{\text{actor}}(\theta; \pi_i^\theta, Q_{\pi_i}^\phi) - \lambda \, \mathcal{L}_{\text{gate}}(\theta; p_i^\theta, p_i^\star) \right).$$

The gate loss $\mathcal{L}_{\text{gate}}$ is a binary cross entropy loss between the predicted gate probabilities $p_i^\theta$ and the probabilities $p_i^\star$ obtained by solving the linear program from Eq. 2.

An illustrative pseudocode description of combining BEXA with SAC is provided in App. A.

## 5. Experiments

To validate our proposed approach, Budgeted EXit Actor (BEXA), and to examine the efficiency of different design choices for employing early exit neural networks (ENNs) within actor-critic methods, we conduct two large-scale experiments based on soft actor-critic (SAC) (Haarnoja et al., 2018) and twin delayed deep deterministic policy gradient (TD3) (Fujimoto et al., 2018).

**Setup and Metrics.** For both SAC and TD3, we refer to their variants with budgeted early exits as BEXA-SAC and BEXA-TD3, respectively. Experiments are conducted on five MuJoCo (Todorov et al., 2012) tasks: Ant, Humanoid, Hopper, Walker2d and HalfCheetah. We report training curves with average return and actor inference speedups measured in floating point operations (FLOPs) and wall-clock time. More details can be found in App. C. Actors and critics are represented by two-layer MLPs with 256 units per layer and ReLU activations, which is a standard architecture for MuJoCo tasks. Larger networks typically do not improve performance in these environments and would artificially favor our method, as relative compute savings increase with network size. For BEXA variants, we add one exit after the first layer, resulting in $K = 2$ exits. For each method-environment pair, we run a hyperparameter sweep over 200 configurations with three seeds and use the same search space across methods where applicable. For the final evaluation, we select the best configurations and evaluate them over 15 seeds, without further tuning. Additional experimental details are provided in App. D.

**Results.** Results can be seen in Fig. 3. BEXA matches or exceeds baselines, with notable gains on Ant, Hopper and Humanoid. We hypothesize that these gains stem from a regularization effect of early exits as actor capacity is reduced. However, we want to emphasize that our goal is not to show large gains in expected return but rather that BEXA can maintain the performance of the base method while *significantly reducing compute*. This is visible as using early-exits can accelerate actor inference tremendously with speedups up to $+584\%$ in FLOPs and $+95\%$ in wall-clock time while sampling in the environment, which accounts for a significant part of deep reinforcement learning (DRL) training. For more complex tasks such as Humanoid, the speedup diminishes as the entire network capacity is required to achieve high performance. However, using a more aggressive budget constraint can yield higher speedups, albeit at the expense of performance. We note, despite introducing new hyperparameters, BEXA required no extra tuning budget relative to its baselines and the budget hyperparameter was straightforward to adapt.

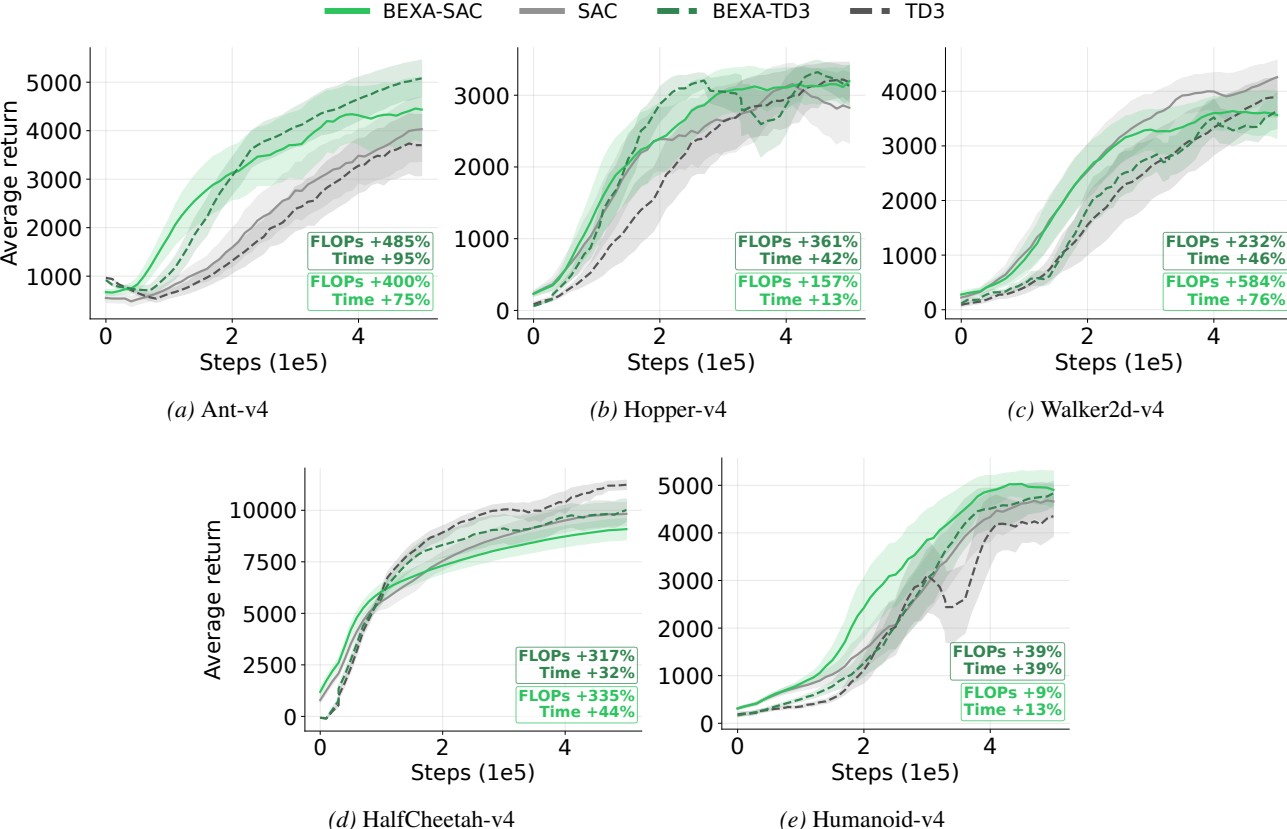

*Figure 3.* Training curves across MuJoCo tasks using 15 seeds. We performed four evaluation runs every 10000 environment steps. Curves are smoothed for readability and shaded with 0.5 standard deviation following Fujimoto et al. (2018). Green annotations indicate the average actor FLOP and wall-clock time speedups of `BEXA` variants relative to their baselines over the *entire* training.

## 5.1. Comparison to Early Exit Alternatives

`BEXA` improves both performance and speed compared to its baseline methods. It still raises the question of how it compares to alternatives in the literature. As discussed in Related Work, direct comparison is difficult as research on early exit networks in DRL is limited, with most of the existing research being conducted in supervised learning. The approach that comes closest is that described in (Kosta et al., 2022), which augments DQN (Hosu & Rebedea, 2016) with early exits, but it targets discrete action spaces, whereas SAC and TD3 use continuous ones. Other DRL acceleration methods, such as quantization and pruning, are orthogonal to our approach and can be combined with it. Benchmarking against these methods offers little insight, especially since methods such as quantization do not reduce FLOPs, but rather the type of operation.

Instead, we propose baselines derived from early exit architectures in supervised learning, adapted to DRL. To our knowledge, these baselines have not been studied in DRL, though they have been effective elsewhere. We compare them in terms of performance, speed-up, and tuning effort.

**Actor Inference.** During sampling in the environment, we already employ the early exits of our actor to achieve speedups during training. Two alternative inference schemes are also worth considering: (i) always use the final (backbone) exit, which often achieves the best performance and (ii) form an ensemble over all exits as in (Sun et al., 2021) leveraging the fact that each branch solves the same task. However, both require full actor inference and thus miss out on acceleration.

**Exit Training.** Instead of using the same loss for every head, we adapt another strategy inspired by self-distillation (Zhang et al., 2019). We train only the final exit (the backbone) with the standard objective and train all earlier exits to imitate its action distribution via an auxiliary imitation loss. This reduces critic complexity, as only one critic is needed for the backbone, but introduces a loss-scale imbalance between the normal loss and the imitation loss, which requires additional hyperparameters.

**Gate Training.** The exit criterion critically affects performance, as it has to reliably pick the best exit while balancing performance and speed for each state. As data sampling and

*Table 1.* Evaluation of alternative components for `BEXA` on MuJoCo using SAC and TD3. We report normalized returns with error bars indicating one standard deviation. For all ablation studies, we use `BEXA` as the base method and replace one component at a time.

| | METHOD | ACTOR SPEEDUP (↑) | | MEAN RETURN (↑) | | BEST RETURN (↑) | |
| --- | --- | --- | --- | --- | --- | --- | --- |
| | | SAC | TD3 | SAC | TD3 | SAC | TD3 |
| ACTOR INFERENCE | BACKBONE ENSEMBLE | $1.0\times$ $1.0\times$ | $1.0\times$ $1.0\times$ | $36.2 \pm 5$ $26.8 \pm 6$ | $33.6 \pm 6$ $32.7 \pm 9$ | $64.1 \pm 16$ $57.7 \pm 22$ | $78.7 \pm 11$ $73.2 \pm 24$ |
| EXIT TRAINING | IMITATE | $1.18\times$ | $1.14\times$ | $29.2 \pm 7$ | $34.0 \pm 10$ | $65.9 \pm 23$ | $71.7 \pm 22$ |
| GATE TRAINING | ADVANTAGE SOFTMAX | $1.164\times$ $1.11\times$ | $1.179\times$ $1.12\times$ | $58.3 \pm 12$ $50.3 \pm 16$ | $40.5 \pm 12$ $\mathbf{49.3 \pm 11}$ | $99.2 \pm 14$ $98.7 \pm 18$ | $79.6 \pm 28$ $\mathbf{88.4 \pm 21}$ |
| TRAIN STRATEGY | STEPWISE | $\mathbf{1.48\times}$ | $\mathbf{1.83\times}$ | $27.8 \pm 4$ | $16.5 \pm 4$ | $40.4 \pm 6$ | $42.4 \pm 23$ |
| | `BEXA` | $1.3\times$ | $1.35\times$ | $\mathbf{62.8 \pm 17}$ | $38.7 \pm 12$ | $\mathbf{101.2 \pm 13}$ | $72.2 \pm 31$ |

learning are tightly coupled, wrong exiting can lead to catastrophic updates. Common heuristics from literature, such as maximum class probability, entropy thresholds and patience are ill-suited as previously discussed due to exploration and smaller model sizes. We consider:

1. *Advantage over backbone*: Taking the exits that have higher value over the backbone. This is similar to the strategy of taking the exit with maximum Q-value (Kosta et al., 2022), but prefers earlier exits.

2. *Softmax over Q-values*: Instead of taking a maximum, we take a softmax over the distribution of Q-values per exit. A temperature hyperparameter controls greediness. This softmax defines the target decision distribution, which we map to gate probabilities via Eq. 1.

Importantly, in App. B we show that our optimal budget-aware exit selection approach allows for direct and intuitive control in the number of FLOPs by selecting an according hard budget constraint. This is significantly harder to achieve with the strategies mentioned above as ablations.

**Training Strategy.** We train all exits and gates simultaneously under a unified objective. Early exit architectures also allow for alternative training schemes. Following (Kosta et al., 2022), we also evaluate a stepwise procedure that sequentially trains each exit branch while freezing the rest of the network, starting with the earliest exit until the final one.

**Setup.** For a fair comparison, all methods were given the same hyperparameter search budget. To better observe the effects of individual components, we drastically reduce network capacity to $4 - 16$ hidden units per layer. The best configurations are re-run to obtain three seeds per setting. Returns are normalized for each environment and then averaged across tasks. We compare speedups of actors in terms

of FLOPs during the whole training. See Table 1 for results.

**Results.** Using alternative actor inference yields no benefit: performance is similar for TD3 and worse for SAC, and it provides no speedup during training unlike the usage of early exits. The imitation-based training objective also underperforms. `BEXA`, which trains all heads using the underlying DRL loss, consistently achieves higher returns. Gate-training results are mixed. As expected, for TD3 we observe higher returns as the greedier gating favors later exits, but at the cost of reducing speedup and making the performance–efficiency trade-off difficult. For SAC, `BEXA` improves both return and speed, suggesting that tighter budget constraints can also act as a form of regularization, boosting performance as well. Lastly, we observe that stepwise training performs poorly. It over-optimized for speedup at the expense of return, and training time increases drastically due to additional gradient steps per iteration. Finally, TD3 and SAC diverge substantially at very low actor capacity, we attribute this to a much narrower hyperparameter region.

## 6. Conclusion

We introduced Budgeted EXit Actor (`BEXA`), a generic method for off-policy actor-critic methods that uses early exits in the actor to reduce the required number of computations under explicit budget constraints. To guarantee that the budget constraints are satisfied, we reformulate the exit selection as a resource allocation problem, which can be efficiently solved using linear programming. `BEXA` is straightforward to tune and matches or even outperforms vanilla baselines and adapted early exit alternatives from the literature across a range of tasks.

**Limitations.** `BEXA` inherits some limitations common to early exit architectures. Training time can increase because all exits must be optimized. To circumvent this, asynchronous training architectures could be used to amortize

such costs by decoupling sampling from learning. Furthermore, dynamic branching makes efficient parallelization on GPUs challenging, a problem that affects the broader early exit community, not just DRL.

**Future Work.** Despite these limitations, BEXA is widely applicable and can be used alongside other acceleration techniques, such as pruning, quantization, and distillation. Future work includes plans to integrate BEXA with additional RL paradigms e.g. model-based RL and scaling to large neural network architectures like ResNets or Transformers (Farebrother et al., 2024), where the additional FLOPs required by the gating mechanism will be negligibly small. In spirit with Sutton's "Bitter Lesson", our aim is to provide general and efficient methods that leverage computation rather than task-specific heuristics, providing a practical foundation for faster and stronger DRL agents.

## Acknowledgments

The authors acknowledge support by the state of Baden-Württemberg through bwHPC and the German Research Foundation (DFG) through grant INST 35/1597-1 FUGG. Experiments were partially conducted on the bwUniCluster 3.0 and the bwForCluster Helix.

## Impact Statement

This paper presents work whose goal is to advance the field of Reinforcement Learning Learning. There are many potential societal consequences of our work, none which we feel must be specifically highlighted here.

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

# A. Pseudocode

In Alg. 1 we provide pseudocode for Budgeted EXit Actor (`BEXA`) using soft actor-critic (SAC) as an example. As stated before, `BEXA` is agnostic with respect to the underlying actor-critic method, which we denote as the base in the algorithm description. The critic updates shown here correspond to those used in SAC.

---

**Algorithm 1** Budgeted EXit Actor (`BEXA`)

---

**Require:** Off-policy base (e.g. SAC or TD3); budget $b$; early-exit actor with exits $i = 1, \ldots, K$; sub-policies $\pi_i(\cdot \mid s)$;
gates $g_i \sim \text{Bernoulli}(p_i^\theta(s))$; critics $Q_i^\phi(s, a)$
Initialize replay buffer $\mathcal{D}$, parameters $\theta, \phi$
**for** environment step $t = 1, 2, \ldots$ **do**
    Observe $s_t$
    **for** $i = 1..K$ **do**
        Compute $p_i^\theta(s_t)$ and sample $g_i \sim \text{Bernoulli}(p_i^\theta(s_t))$
        **if** $g_i = 1$ **then**
            $a_t \sim \pi_i(\cdot \mid s_t)$; **break**
        **end if**
    **end for**
    Step environment, observe $(r_t, s_{t+1}, d_t)$
    Store $(s_t, a_t, r_t, s_{t+1}, d_t)$ in $\mathcal{D}$
    **for** update step $u = 1, \ldots, U$ **do**
        Sample minibatch $\mathcal{B} \subset \mathcal{D}$
        **(1) Critic update (base-agnostic).** For each exit $i = 1..K$:
        Compute a TD target $y_i^{\text{(base)}}$ per the chosen off-policy base, e.g. for SAC:

$$y_i^{\text{(SAC)}} = r + \gamma(1-d) \, \mathbb{E}_{a' \sim \pi_i(\cdot \mid s')} \Big[ \min_{m \in \{1,2\}} Q_{i,m}^{\bar{\phi}_m}(s', a') - \lambda \log \pi_i(a' \mid s') \Big].$$

        Then update $\phi$ by a gradient step on $\frac{1}{|\mathcal{B}|} \sum (Q_i^\phi(s, a) - y_i^{\text{(base)}})^2$.
        **(2) Linear program for exit mixture.**

$$\alpha^\star = \arg\max_{\alpha \in \mathbb{R}^K} v^\top \alpha \quad \text{s.t.} \quad c^\top \alpha \leq b, \ \ \alpha \geq 0, \ \ \mathbf{1}^\top \alpha = 1$$

        **(3) Map mixture to target gate probabilities.**
        Using Eq. 1 to compute $p^\star$ recursively from $\alpha^\star$:

$$p_1^\star(s) = \alpha_1^\star(s), \qquad p_i^\star(s) = \frac{\alpha_i^\star(s)}{\prod_{j<i}(1 - p_j^\star(s))} \ \ \text{for } i = 2, \ldots, K.$$

        **(4) Actor update (`BEXA` objective).**
        $\theta \leftarrow \theta + \eta_\pi \nabla_\theta \frac{1}{|\mathcal{B}|} \sum_{s \in \mathcal{B}} J_{\text{BEXA}}(\theta; s, p^\star)$
    **end for**
**end for**

---

## B. Effect of the Budget on Computational Cost and Return

Here, we investigate how we can control the numbers of required floating point operations (FLOPs) using our resource allocation formulation. In Fig. 4 we see how the expected FLOPs linear scale with the normalized budget. Furthermore, Fig. 5 highlights that the performance increases with the allowed budget.

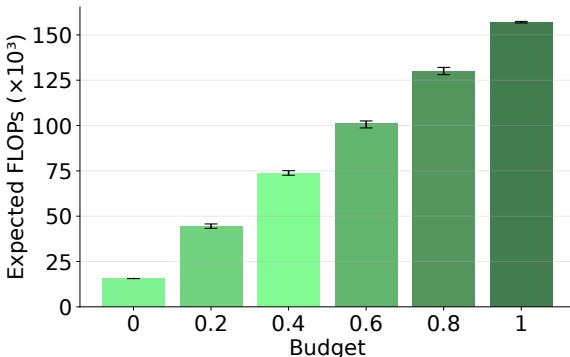

*Figure 4.* Average FLOPs for the actor in relation of budget $b$ when using `BEXA-SAC`. Evaluated on the Halfcheetah-v4 environment using $\sim 70$ runs per bar. One standard deviation is plotted. This shows that budget regulates flops explicitly and in a intuitive way.

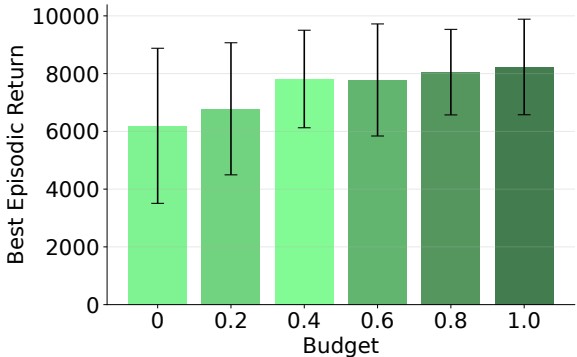

*Figure 5.* Best Return reported for the actor in relation of budget $b$ when using `BEXA-SAC`. Evaluated on the Halfcheetah-v4 environment using $\sim 70$ runs per bar. One standard deviation is plotted. Giving more budget allows for higher return.

## C. Early-Exit Inference Implementation

The main implementation of our early exit neural networks is written in PyTorch (Ansel et al., 2024). However, measuring inference speed in this framework is challenging for small networks, since framework overhead such as Python to C++ dispatch and kernel launch overhead can start to dominate the overall inference time. Consequently, we were not seeing the same significant gains in wall-clock time as we would theoretically expect from the reduction in FLOPs.

To address this issue and more accurately assess the effectiveness of our method, we also implemented the early exit neural networks in the Rust programming language using the Burn crate (Simard et al., 2026). Framework overhead is much smaller there, and only much larger networks might be slower. We evaluated both implementations on the HalfCheetah task, measuring inference speed of an early exit policy under different budget constraints. To ensure a fair comparison, we also evaluated a static baseline without any gates, implemented in the respective framework. The results can be seen in Table 2.

The Rust implementation improves inference speed for most budget settings, achieving a speedup of up to $1.91\times$. However, with the full budget, it is slightly slower, as the gating mechanism introduces some overhead compared to a network without any early exits. In contrast, the PyTorch implementation is strongly affected by framework overhead and cannot demonstrate

*Table 2.* Wall-clock time comparison of the early exit policies for different budgets and implementations. We sampled 2048 random states from the HalfCheetah task and measured inference time over 20 trials.

| Budget | Exp. FLOPs | FLOP Speedup | PyTorch ns/input | PyTorch Speedup | Rust ns/input | Rust Speedup |
|--------|-----------|-------------|-----------------|----------------|--------------|-------------|
| 0.0 | 15,168 | 9.65× | $44,900 \pm 646$ | $0.93 \pm 0.03\times$ | $5,533 \pm 76$ | $1.91 \pm 0.14\times$ |
| 0.2 | 41,459 | 3.53× | $48,217 \pm 1,053$ | $0.87 \pm 0.03\times$ | $6,664 \pm 160$ | $1.58 \pm 0.10\times$ |
| 0.4 | 67,686 | 2.16× | $51,926 \pm 2,050$ | $0.81 \pm 0.03\times$ | $7,958 \pm 295$ | $1.33 \pm 0.10\times$ |
| 0.6 | 93,978 | 1.56× | $54,667 \pm 1,466$ | $0.77 \pm 0.03\times$ | $8,927 \pm 443$ | $1.18 \pm 0.08\times$ |
| 0.8 | 120,205 | 1.22× | $57,813 \pm 1,285$ | $0.73 \pm 0.03\times$ | $9,897 \pm 494$ | $1.07 \pm 0.08\times$ |
| 1.0 | 146,496 | 0.99× | $61,394 \pm 2,115$ | $0.68 \pm 0.02\times$ | $11,165 \pm 768$ | $0.95 \pm 0.09\times$ |

the true potential of early-exit inference for such small networks. Thus, throughout our paper, we report Rust-based inference times.

# D. Hyperparameters

To follow best practices (Eimer et al., 2023), we report all the hyperparameters that were relevant to our experiments, along with the search spaces that were used for each one. These exploratory searches are not included in the paper. Based on the outcomes of these searches, we identified the most promising bounds and fixed them uniformly across all methods. We then performed a final hyperparameter search within a fixed computational budget, the results of which are reported in the paper. Hyperparameters were tuned using random search. For continuous hyperparameters, we used q-log-uniform, which samples logarithmically and rounds to discrete multiples of a step $q$.

In Tab. 3 and Tab. 4 we highlight the search spaces used for Fig. 3. For the ablation studies presented in Tab. 1, we used the search spaces in Tab. 6 and Tab. 7. Furthermore, to facilitate comparison of performance across environments, we normalize the return when aggregating results, see Tab. 5

*Table 3.* Hyperparameter configuration used for comparison of SAC and `BEXA`-SAC.

| Hyperparameter | Values / Range |
|----------------|----------------|
| `batch_size` | 256 |
| `learning_starts` | 5000 |
| `policy_frequency` | 2 |
| `autotune` | True |
| `gamma` | 0.99 |
| `tau` | q-log-uniform (min: 1e−3, max: 1e−2, q: 1e−3) |
| `policy_lr` | q-log-uniform (min: 1e−4, max: 7e−4, q: 1e−4) |
| `q_lr` | q-log-uniform (min: 2e−4, max: 1e−3, q: 1e−4) |
| `budget` | [0.2, 0.3, 0.4, 0.5, 0.6, 0.7, 0.8, 0.9, 1.0] |
| `actor_inference` | early_exit |
| `critic_kind` | multi_head |
| `actor_training` | all_exits |
| `gate_training` | budget |
| `training_scheme` | jointly |
| `total_timesteps` | 500000 |

*Table 4.* Hyperparameter configuration used for comparison of TD3 and `BEXA`-TD3.

| Hyperparameter | Values / Range |
|---|---|
| batch_size | 256 |
| learning_starts | 25000 |
| policy_frequency | 2 |
| gamma | 0.99 |
| tau | q-log-uniform (min: $1e-3$, max: $1e-2$, q: $1e-3$) |
| lr | q-log-uniform (min: $1e-4$, max: $1e-3$, q: $1e-4$) |
| policy_noise | 0.2 |
| exploration_noise | 0.1 |
| noise_clip | 0.5 |
| budget | [0.2, 0.3, 0.4, 0.5, 0.6, 0.7, 0.8] |
| actor_inference | early_exit |
| critic_kind | multi_head |
| actor_training | all_exits |
| gate_training | budget |
| training_scheme | jointly |
| total_timesteps | 500000 |

*Table 5.* Normalization constants used to scale returns for MuJoCo tasks.

| Environment | Normalization (return) |
|---|---|
| HalfCheetah-v4 | 60.0 |
| Walker2d-v4 | 30.0 |
| Hopper-v4 | 30.0 |
| Humanoid-v4 | 50.0 |
| Ant-v4 | 40.0 |

*Table 6.* Sweep configuration for BEXA-SAC and alternative ablation components.

| Hyperparameter | Values / Range |
|---|---|
| batch_size | 256 |
| hidden_size | [4, 8, 16] |
| learning_starts | 5000 |
| policy_frequency | 2 |
| autotune | True |
| gamma | 0.99 |
| tau | q-log-uniform (min: $1e-3$, max: $1e-2$, q: $1e-3$) |
| policy_lr | q-log-uniform (min: $1e-4$, max: $7e-4$, q: $1e-4$) |
| q_lr | q-log-uniform (min: $3e-4$, max: $1e-3$, q: $1e-4$) |
| imitate_loss_scale | q-log-uniform (min: $1e-2$, max: $4e-1$, q: $1e-2$) |
| budget | [0.1, 0.2, 0.3, 0.4, 0.5, 0.6, 0.7, 0.8, 0.9] |
| gate_softmax_tmp | q-log-uniform (min: $1e-1$, max: $2.0$, q: $1e-1$) |
| actor_inference | [backbone, ensemble] |
| actor_training | [imitate, all_exits] |
| gate_training | [budget, adv, softmax] |
| training_scheme | [stepwise, jointly] |
| total_timesteps | 500000 |

*Table 7.* Sweep configuration for BEXA-TD3 and alternative ablation components.

| Hyperparameter | Values / Range |
|---|---|
| batch_size | 256 |
| hidden_size | [4, 8, 16] |
| learning_starts | 25000 |
| policy_frequency | 2 |
| gamma | 0.99 |
| tau | q-log-uniform (min: $1e-3$, max: $1e-2$, q: $1e-3$) |
| lr | q-log-uniform (min: $1e-4$, max: $1e-3$, q: $1e-4$) |
| policy_noise | [0.1, 0.2, 0.3, 0.4] |
| exploration_noise | [0.1, 0.2, 0.3] |
| noise_clip | [0.1, 0.2, 0.3] |
| imitate_loss_scale | q-log-uniform (min: $1e-2$, max: $4e-1$, q: $1e-2$) |
| budget | [0.1, 0.2, 0.3, 0.4, 0.5, 0.6, 0.7, 0.8, 0.9] |
| gate_softmax_tmp | q-log-uniform (min: $1e-1$, max: $2.0$, q: $1e-1$) |
| kl_eps | q-log-uniform (min: $1e-1$, max: $2.0$, q: $1e-1$) |
| actor_inference | [backbone, ensemble] |
| actor_training | [imitate, all_exits] |
| gate_training | [budget, adv, softmax] |
| training_scheme | [stepwise, jointly] |
| total_timesteps | 500000 |

