# OpenReview forum: "Mind the Budget: Accelerating Deep Reinforcement Learning using Constrained Early Exit Neural Networks"
_ICML.cc/2026/Conference — ICML 2026 regular_

### Official Review · Reviewer_neN3 · 2026-02-16

**Soundness:** 2
**Presentation:** 3
**Significance:** 2
**Originality:** 2
**Overall Recommendation:** 3
**Confidence:** 4

**Summary:**

The paper proposes „BEXA“ (Budgeted EXit Actor), which allows to make use of Early Exit Neural Networks (ENN) to boost inference time in deep RL. The ENN provides several exits and refines the policy towards the end. However, while the policy provides the best actions at the latest exit it also takes highest compute. This is why early exits in the neural network provide actions/policies that only take little compute. The authors provide a formal introduction of the constrained constrained allocation problem and solve it (i.e., the gate activations) using an LP that is solved per batch (in addition to gradient updates to optimize the policies). On a set of Mujoco benchmark they show that the method performs well in term of achievable return despite the fact of being computationally lightweight.

**Compliance With Llm Reviewing Policy:**

Affirmed.

**Final Justification:**

The authors answered the questions and I have some concerns left regarding the experimental results; However, those not necessarily should prevent this paper from being accepted. I have no problem with this paper being accepted.

**Key Questions For Authors:**

- Can you provide an analysis on wall-clock inference time?
- In the introduction you say “Typically, early exit branches are trained directly on fixed ground truth data, whereas in RL the behavior of the agent changes over time. In addition, the predicted actions influence the state distribution encountered by the agent. Poorly chosen actions of ENNs can therefore lead to learning instabilities.” – can you please elaborate a bit more, what’s so special about RL here?
- “Preliminary experiments indicated that maintaining a separate critic per exit substantially improved learning stability.” – Can you elaborate why this is?
- Why does BEXA outperform the baselines with respect to achievable return? This is surprising.

**Limitations:**

Yes, I think so.

**Strengths And Weaknesses:**

**Strengths:**
- The paper is well written and easy to understand
- The proposed method is simple yet effective. It can be used on a variety of RL algorithms and makes only mild assumptions to the overall framework
- The LP allows to explicitly “configure” the average inference latency (per batch) instead of  only “steering” it towards some direction (but I also have a weakness wrt to that feature below)

**Weaknesses:**
- The inference is configured per batch, however, the paper motivates inference latency to be critical for control applications e.g. in robotics. Here, however, we would like to have a deterministic and low inference latency for any state input – while the paper constrains the inference latency on batch level. The authors should invest some more time on the actual application scenario that benefit from the proposed method.
-	Following on the previous comment I like the illustration using chess in the motivation. While I think the overall problem setting is a better fit to the proposed method (in chess we want to constrain the *overall* inference latency over all the actions I take during one game) I miss such experiments in the paper that study those problems.
-	I miss a study that analyzes wall-clock time during training and inference. I have read the arguments about not having it analyzed but I don’t buy it. All the pipelines are equally well (or badly) optimized. In addition, I understand the authors apply an online RL setting (forward pass, solving LP, backward pass) which is also a maybe not the best setup to evaluate the proposed methods
-	Experimental settings: (1) The used neural network configurations for the actors are surprisingly small. (2) The used seeds (i.e., 3 for the top-5) are too low. (3) What about actual training time increase of BEXA? (4) Ablation: training/inference time gains vs. loss in achievable return is missing

**Minor comments:**
- Line 070: I don’t see “extensive ablation studies” in the paper
- Section 2.2, lines 108-121 can be omitted
- Section 4 line 152: for integrating “an” early exit neural network…
- Line 169: the stochastic gating encourage exploration – not in a sense that we use the term “exploration” in RL. Maybe you should re-formulate this
- The LP uses a scalar budget $b$. An ablation of $b$ would also be interesting
- “quantization do not reduce FLOPs, but rather the type of operation.” Of course it can when applied in a structural manner (which is what you should opt for). Please re-formulate.
- The paper would benefit from experiments that require more cmplex and rich NN structures (e.g. vision) to prove its benefits.

---

> ### Author Rebuttal · Authors · 2026-03-31
>
> Dear Reviewer,
> We thank you for the detailed and thoughtful feedback. We appreciate the critical perspective and believe that several of the points raised helped clarify important aspects of our work. We address them below:
>
> **1. Batch vs. per-state inference and applicability to real-time settings**
>
> We would like to clarify that BEXA operates **per state**, not per batch.
>
> While LPs are solved in a batched manner for efficiency, each LP corresponds to an individual state. This allows us to guarantee that the compute budget is satisfied on a per-state basis in expectation. We agree that the run time is stochastic, thus our approach is not feasible if we need to guarantee that we are below the budget constraint for each inference separately. Still, having a faster average inference can result in a faster average control frequency, which can in a lot of scenarios lead to better final performance. This can be also beneficial even in the robotic setting, but also in the context of hosting large language models (LLMs)  [1].
>
> **2. Wall-clock time analysis**
>
> It is common in early exit network or sparse neural network pruning research to measure the performance in FLOPs. Still, we agree that wall-clock time is an important metric. However, in practice, it is difficult to measure fairly for early-exit architectures due to limitations of current software stacks. Our implementation is based on PyTorch, where dynamic control flow introduces several sources of overhead:
> - Python-C++ overhead when evaluating exit conditions,
> - kernel launch overhead for small branches, which can dominate compute,
> - lack of efficient support for dynamic or conditional execution in current libraries (e.g., oneDNN).
> Although early-exit branches are significantly smaller than the main layers, their runtime cost is often comparable due to substantial overhead. For example, a small layer (e.g. 256 -> 5) does not execute significantly faster than a larger layer (e.g. 256 -> 256), as kernel launch and control overhead dominate. We implemented a C++ version to reduce overhead and observed preliminary speedups (~30% on CPU for Hopper), with further improvements achievable through more specialized optimizations. Overall, current ML frameworks are primarily optimized for large, static networks, whereas our smaller early-exit models introduce dynamic execution patterns that are not yet well supported. We view this as an important systems challenge, but one that is beyond the scope of this work.
>
> **3. Online RL setting and evaluation**
>
> The online RL setting is particularly relevant because sampling cost can play a major role. In many RL setups, sampling and environment interaction dominate overall runtime. Our method reduces the cost of sampling by enabling early exits, while the additional LP cost remains small, as discussed in the paper.
>
> While we follow the standard SAC/TD3 setup (one sample per update), in scenarios with higher sampling throughput, the benefits of reduced inference cost can become even more pronounced.
>
> **4. Experimental Setup**
>
> The chosen architectures are standard for MuJoCo. Larger networks typically do not improve performance in these tasks and would artificially favor our method, since relative compute savings would increase. We therefore opted for a fair and commonly used setup.
>
> We agree that more seeds would strengthen the evaluation. For the final version, we plan to re-evaluate with a larger number of seeds to improve robustness.
>
> Solving the LP introduces additional cost during training. However, in our setting, training is dominated by actor-critic updates. Thus overall training time increases. The additional cost can be amortized in settings where sampling is more frequent or more expensive.
> We agree this is an important perspective. However, as discussed above, wall-clock comparisons are currently difficult to evaluate fairly due to missing support for fast small ENNs.
>
>
> **Questions:**
>
> **Q1.**
> Please see our comment on Wall-clock time analysis
>
> **Q2.**
> In RL, early approximation errors can affect the data distribution through the policies actions, potentially leading to instability or divergence. This makes training early-exit architectures harder than in supervised settings.
>
> **Q3.**
> Using a single critic for all exits led to instability, as changes in gating behavior can rapidly change the overall policy. Learning a separate critic per exit stabilizes training by decoupling these effects.
>
> **Q4.**
> This is consistent with prior work on early-exit networks, see [2]. Early exits can act as a form of regularization, encouraging simpler representations and sometimes improving performance. However, our primary goal is to maintain performance under reduced compute.
> [1] Yue et al. “DeeR-VLA: Dynamic Inference of Multimodal Large Language Models for Efficient Robot Execution” 2024.
> [2] Rahmath et al. "Early-Exit Deep Neural Network: A Comprehensive Survey." 2024.

---

> > ### Author Rebuttal · Reviewer_neN3 · 2026-04-01
> >
> > > 1. Batch vs. per-state inference and applicability to real-time settings
> >
> > Thanks, that really resolved my issue on this point. Satisfying compute budget per state in estimation is totally fine.
> >
> > > 2. Wall-clock time analysis
> >
> > In understand the technical arguments - there are no efficient implementation available that makes use of the sparse nature of small early-exit-networks in order to leverage their full latency benefit. However, this still raises the question what performance benefit one might actually have from using BEXA in practice. This is - at least to me - not satisfying.
> >
> > > 3. Online RL setting and evaluation
> >
> > Partially resolved. However, I still might be more interested in the performance gains I will observe under an inference or a deployment setting in particular. How are the benefits in training compared to inference-only (greedy) settings?
> >
> > > 4. Experimental Setup
> >
> > My questions here are partially answered, however, I was expecting more ablations already at submission time (as I have some concerns that I cannot judge at this moment). Also reading through the other reviews leave me a bit skeptical at this point.
> >
> > In the light of this author response I raise my score to 3.

---

> > > ### Author Response · Authors · 2026-04-07
> > >
> > > Dear Reviewer,
> > >
> > > We thank you for the follow-up response. We want to address these points:
> > >
> > >
> > > **Wall-clock time analysis**
> > >
> > > Regarding wall-clock time, we have now implemented our early-exit networks fully in Rust using the Burn/Candle backend. In contrast to PyTorch, which typically relies on MKL or oneDNN, this implementation uses a Rust-native general matrix multiply (GEMM) backend with lower dispatch costs and less framework overhead. While this backend is slower for large networks, it is more efficient for the small networks used in our early-exit branches. We do not view this as a final optimized system, but rather as a proof of concept showing that even with current tools, early-exit networks can already provide real latency benefits in practice. In the future, more specialized implementations could likely yield even larger speedups.
> > >
> > > We evaluated this implementation on HalfCheetah under different budget constraints, using trained policies and $20$ trials over $2048$ sampled states. For each budget, we report not only the expected FLOPs and FLOP speed up as we did in our paper, but also the measured inference time of both our PyTorch/C++ implementation and our Rust/Burn implementation, each compared against a fair static-network baseline implemented in the same framework.
> > >
> > > | Budget | Expected FLOPs | FLOPs Speedup | PyTorch ns/input | PyTorch Speedup | Burn ns/input | Burn Speedup |
> > > |---|---|---|---|---|---|---|
> > > | 0.0 | 15,168 | 9.65x | 44,900 ± 646 | 0.93 ± 0.03x | 5,533 ± 76 | 1.91 ± 0.14x |
> > > | 0.2 | 41,459 | 3.53x | 48,217 ± 1,053 | 0.87 ± 0.03x | 6,664 ± 160 | 1.58 ± 0.10x |
> > > | 0.4 | 67,686 | 2.16x | 51,926 ± 2,050 | 0.81 ± 0.03x | 7,958 ± 295 | 1.33 ± 0.10x |
> > > | 0.6 | 93,978 | 1.56x | 54,667 ± 1,466 | 0.77 ± 0.03x | 8,927 ± 443 | 1.18 ± 0.08x |
> > > | 0.8 | 120,205 | 1.22x | 57,813 ± 1,285 | 0.73 ± 0.03x | 9,897 ± 494 | 1.07 ± 0.08x |
> > > | 1.0 | 146,496 | 0.99x | 61,394 ± 2,115 | 0.68 ± 0.02x | 11,165 ± 768 | 0.95 ± 0.09x |
> > >
> > > These results show that our Rust implementation achieves clear speedups relative to a static Rust baseline for most budget settings. Only at the full budget, $b=1.0$, it becomes slightly slower, since the gating mechanism still introduces overhead compared to a network without any early-exits. By contrast, the PyTorch/C++ implementation is heavily affected by framework overhead, especially in the execution of the gates, and therefore does not yet reflect the true potential of early-exit inference. We will release these implementations to support further analysis and reproducibility. We also emphasize that these measurements reflect pure inference only.
> > >
> > > **Online RL setting and evaluation**
> > >
> > > Concerning training versus inference-only deployment, we would like to stress that training the network jointly with its gates and exits is important in its own right. When exits are incorporated already during training, the policy can adapt to them and exploit additional signals, such as the Q-values associated with different exits. This allows the model to learn solutions that are aware of the early-exit structure from the beginning learning optimal behavior under budget constraints, rather than adding exits only after training. For this reason, we believe there is clear value in integrating exits during training, not only at deployment time.
> > >
> > > **Experimental Setup**
> > >
> > > For the final version, we plan to reevaluate our experiments with the new Rust implementation using more seeds, and report both full training-time and inference-time measurements.

---

### Official Review · Reviewer_M5NT · 2026-03-11

**Soundness:** 3
**Presentation:** 2
**Significance:** 3
**Originality:** 3
**Overall Recommendation:** 4
**Confidence:** 4

**Summary:**

The work extends the idea of an early exit neural network (ENN) to off-policy actor-critic methods, aiming to speedup inferences. They apply the idea only to the actor but not to the critics. The design requires balancing performance and the cost of inference (measured by FLOPs), which is reflected by the probability of exiting at earlier gates of the actor network. The parameters of the policy and the gates are jointly optimized, where the sub-policies are trained with corresponding critics, and the gates are trained with the closed-form solutions given by LP. They conduct experiments on five MuJoCo tasks, indicating comparable performance and improved FLOPs compared to the baseline actor-critic methods they are building on.

**Compliance With Llm Reviewing Policy:**

Affirmed.

**Key Questions For Authors:**

Please have a look at a few questions already raised above.
Another question: the sub-policies $(\pi_i)_i$ overlap in the earlier layers. Does this fact (earlier layers contribute more) affect training?

**Limitations:**

Yes

**Strengths And Weaknesses:**

## Strengths
- The proposed method is technically sound, and, to the best of my knowledge, novel. They extended the idea of early exit NN, which has been adopted in the context of CV and NLP to show advantages, to off-policy actor-critic. The optimization of the probability of exiting from earlier gates requires a non-trivial design. While they use classic actor-critic methods to learn each sub-policy (policies defined by early exiting), they figured out the closed-form solution of the probability distribution, given the sub-policies and the costs, as a signal to guide learning of the gates.
- Computational cost has been an issue in modern RL. Hence, improving computation provides important values to the field.
- The clarity is pretty ok in general. However, there are a few parts that can be improved, which I will list later.

## Weaknesses
### Presentation of a few parts
- Optimal Budget-Aware Exit Selection: You somehow abruptly switch from state-dependent notations to a more general notation (without states being specified). It is fine to do so to focus on introducing the method to solve $\alpha$. However, it’s perhaps better to add some words for a smoother transition. Otherwise, I was confused at first glance, and only realized it later that the $\alpha^*$ is a state-dependent function. Also, denoting it as $\alpha^\*$ can make one think this is the universally optimal solution; however, if I understand correctly, this is just the optimal given the current policy (network). Maybe you can also clarify this in your presentation.

- The BEXA Training Objective: I understand that the actor network involves two parts: the class “layers part” and the “gates part”, where you learn the layers using the classic actor-critic methods and learn the gates using the loss between the LP solution (given the layers) and the current probability distribution. Continuing a bit on the above point, such $p^\*$ is the solution given the layers, meaning that there is an “order”. I wonder, do you / can you learn both the layers and the gates jointly?

- The notation $Q_{i,m}^{\bar\phi_m}$ in the pseudocode is a bit arbitrary without a proper description. People with enough background will know what you mean, but this is not understandable in general.

### Others
- I am not sure I understand “To minimize the effect of seed variance, we re-evaluate the top 5 runs per sweep with two additional seeds, giving us three seeds in total, and select the best by mean return over the entire training.” Also, in Figure 3, you mention you only do 1 evaluation run per 10k steps. - I don’t normally complain about repetitions, but this total number of runs and the selection do not seem very natural to me. As you already have pretty few seeds, I would suggest not having too few evaluation runs as well. At the moment, the learning curves do not look very informative.

## Comment
This is just a comment, but not a complaint. Seeing an effect already appears with a two-layer MLP (with 3 exits) is pretty interesting. It will perhaps be even more interesting to see how such a design works with deeper networks.

## Minor
- Typo: toward the end of page 4, $c_n$ should perhaps be $c_K$.

- I might have overlooked, but did you mention how to choose lambda, the balance between two losses?

---

> ### Author Rebuttal · Authors · 2026-03-31
>
> Dear Reviewer,
> We thank the reviewer for the detailed and constructive feedback, as well as for recognizing the novelty and motivation of our work. We address the points below:
>
> 1. Optimal Budget-Aware Exit Selection (notation and interpretation of $\alpha$)
> We appreciate this comment and agree that the transition in notation could be clearer. In the current version, we drop the explicit state-dependence of $\alpha$ for conciseness. However, we acknowledge that this may cause confusion. In the final version, we will make this dependence explicit (e.g., $\alpha_s$) and add clarifying text to ensure a smoother transition.
>
> We also agree that the notation $\alpha^*$ may suggest a globally optimal solution. To clarify, $\alpha^*$ is only optimal given the current network parameters. We will revise the presentation to make this dependency explicit.
>
> 2. BEXA training objective and joint optimization
>
> Yes, both the network layers (sub-policies) and the gating mechanism are trained jointly. This is consistent with standard practice in early-exit literature, where joint training enables shared representations across exits. While the LP solution provides a target for the gating distribution given the current sub-policies, both components are optimized simultaneously during training.
>
> We also explored alternative training strategies, such as:
> - sequential stepwise training (which is included in our analysis)
> - alternating updates between gates and sub-policies.
>
> These approaches led to slower training and worse performance, without improving returns. We can include the results of alternating updates in the appendix in the final version.
>
> 3. Notation of Q in the pseudocode
>
> We agree that the pseudocode can be made clearer. In the final version, we will revise the notation and provide a clearer description of the $Q$-function in the clipped double Q-learning to improve readability.
>
> 4. Evaluation protocol and number of runs
>
> We understand the concern and will clarify this part of the paper. Our evaluation protocol (one evaluation run every 10k steps) follows common practice (e.g., Soft Actor-Critic evaluates in the same way). Our goal is to demonstrate that performance is maintained under reduced compute, rather than improved.
>
> Regarding seed selection: we first perform a hyperparameter sweep, then re-evaluate the top configurations with additional seeds to reduce variance. We will revise the description to make this process more transparent and additionally reevaluate the best runs with more seeds.
>
> 5. Comment on shallow networks
>
> We thank the reviewer for this observation. We agree that it is interesting that the effect already appears in relatively small networks. Extending the approach to deeper architectures is a natural and promising direction for future work.
>
> 6. Minor points
>
> - Thank you for pointing out the typo. We will fix it in the final version.
> - The balance parameter $\lambda$ is selected via hyperparameter tuning. This parameter is standard in early-exit architectures and controls the trade-off between losses across exits.
>
> We thank the reviewer again for the helpful feedback and will incorporate these improvements in the final version.

---

> > ### Author Rebuttal · Reviewer_M5NT · 2026-04-02
> >
> > My concerns are mostly addressed. At the same time, I understand and agree with other reviewers that there are a few aspects worth reporting. 1. Increase the number of seeds, which the authors have promised. 2. I understand the authors' concern about reporting wall-clock time. However, I personally think it's still worth reporting to serve as a reference and discussing it, rather than omitting it directly. I have decided to keep my score.

---

### Official Review · Reviewer_tWHg · 2026-03-13

**Soundness:** 3
**Presentation:** 4
**Significance:** 3
**Originality:** 2
**Overall Recommendation:** 5
**Confidence:** 4

**Summary:**

The authors show how early exit neural networks can be applied inside RL tasks. While ENNs have previously been used in Q-learning, the authors show how they can be used inside the actor-critic framework and demonstrate that it can be jointly trained with exits to lower the training budget. They introduce BEXA and its training objective.

Experiments on a number of standard RL benchmarks are presented

**Compliance With Llm Reviewing Policy:**

Affirmed.

**Final Justification:**

I have no further comments and still am in favor of acceptance

**Key Questions For Authors:**

1) There's been a bit of recent work on using MoEs to sclae deep RL. Would it be possible to combine BEXA into the router layer of an MoE for early exits?

2) Is there any results showing when the early exits occur? Would be interesting to see which states make it deeper into the network and if there's any correlation between exit and the "obviousness" of the solution in a particular state

**Limitations:**

Yes

**Strengths And Weaknesses:**

Strengths

The paper has a clearly defined goal (using ENNs in actor-critic) and does a good job addressing it. The architecture is very clearly motivated by the goal.

Treating the exit selection problem as a budget-constrained optimization problem is a novel contribution and gives a good theoretical basis for the BEXA objective.

The algorithm presented is compatible with standard RL algorithms and seems to boost performance

Paper is very well written

Weaknesses

Limited novelty in the architecture, but the overall framework makes up for this

Speedups are mostly realized in FLOPs and not wallclock time due to GPUs not being particularly suitable to benefit from ENN efficiencies

Evaluation domain can be expanded beyond just MuJoCo

A more detailed hyperparameter testing would be appreciated

---

> ### Author Rebuttal · Authors · 2026-03-31
>
> Dear Reviewer,
> We thank you for the positive feedback and for highlighting the clarity, motivation, and theoretical grounding of our approach. We address the questions below:
>
> **Questions:**
> **Q1. Combining BEXA with MoE routing**
>
> Yes, we believe this is a promising direction. In principle, BEXA can be integrated into a mixture-of-experts (MoE) setting, where different experts may have varying computational costs. In such a scenario, BEXA could be used to learn a budget-aware routing policy, selecting experts in a way that optimizes the trade-off between performance and computation. This would naturally extend our formulation beyond early exits within a single network to adaptive routing across multiple experts, potentially even across different policies. We see this as an interesting avenue for future work.
>
> **Q2. When do early exits occur?**
>
> We agree that analyzing when early exits occur and how they relate to state “difficulty” or “obviousness” is an interesting question. While we do not include this analysis in the current version of the paper, we believe it would be straightforward to add and would provide additional insight. We would be glad to include such an analysis in a final version, as this was also requested by some other reviewers.

---

> > ### Author Rebuttal · Reviewer_tWHg · 2026-03-31
> >
> > Thank you for your response, my questions have been answered

---

### Official Review · Reviewer_yrTV · 2026-03-19

**Soundness:** 3
**Presentation:** 2
**Significance:** 2
**Originality:** 2
**Overall Recommendation:** 4
**Confidence:** 3

**Summary:**

Designing agents that can reason about their computation is important since not all inputs to the network require the same amount of computation. This paper focuses on this problem from the angle of early-exit neural networks (ENNs). It implements ENN's ideas into the actor of two off-policy continuous control deep RL methods. Each gate controlling early exist at each layer is trained using a supervised loss based on a simple linear program that maximizes the action value of each conditioned on each exist while respecting the compute budget. The results show that the proposed methods can achieve the same level of performance under a smaller compute budget, measured by FLOPs.

**Compliance With Llm Reviewing Policy:**

Affirmed.

**Key Questions For Authors:**

- Could the budget-aware gating mechanism instead be formulated as an RL problem, where an internal controller decides which exit to use in order to optimize both return and computation cost? How do the authors expect such an approach to compare with the linear-program-based formulation?
- What properties of the state make early exits preferable to late exits, or vice versa? Can the authors provide empirical analysis to clarify this?
- How should one think about computation allocation over the full trajectory, rather than on a per-state basis?

**Limitations:**

The authors mention the limitations of their work.

**Strengths And Weaknesses:**

**Strengths:**
- The paper is well motivated, clearly written, and easy to follow.
- The budget-aware gating formulation is simple, intuitive, and technically reasonable.
- The broader problem of building agents that can adapt their computation to the demands of the input is important and timely.


**Weaknesses**
- The evaluation scope is limited. The paper considers only five MuJoCo environments, which makes it difficult to assess the generality of the conclusions.
- The empirical evidence for improvement is not fully convincing. The results are based on only five seeds, and many of the learning curves show substantial overlap in their confidence intervals. As a result, the claimed gains do not appear especially robust.
- The paper does not analyze several important aspects of the problem. First, under a fixed total compute budget over a long interaction horizon, it is unclear how BEXA should optimally allocate computation across states and time. Second, the paper does not provide much insight into what properties of the state lead the policy to favor earlier versus later exits. This interpretability question seems central to the problem setting.
- The method appears potentially sensitive to hyperparameters. The authors report trying 200 hyperparameter configurations, but the paper does not clearly analyze which hyperparameters matter most or how stable the method is across settings.
- The trade-off between compute budget and performance is not discussed deeply enough. This trade-off is central to the paper’s motivation, yet the presentation remains mostly descriptive rather than analytical.

I think the paper addresses an important research direction: how to build agents that reason about their computation usage. The idea is interesting, and the paper is clearly written. However, I do not find the empirical results convincing enough for acceptance. The evaluation is limited to a small set of MuJoCo tasks, the reported gains do not appear robust, and several central questions, such as compute allocation over time, sensitivity to hyperparameters, and the characteristics of states that trigger early versus late exits, remain insufficiently explored. Overall, while I find the direction promising and would like to eventually see this paper accepted, I do not think the current manuscript is ready for acceptance.

---

> ### Author Rebuttal · Authors · 2026-03-31
>
> Dear Reviewer,
> We thank you for the thoughtful and constructive feedback. Below, we provide our view to your main concerns:
>
> **1. Limited evaluation scope (MuJoCo only)**
>
> Our goal is to validate the *core principle* of budget-aware early-exit policies in RL. For this, we evaluated across all standard MuJoCo benchmarks covering different dynamics and control challenges. Importantly, early-exit neural networks (ENNs) have been extensively studied in other domains (e.g., vision and NLP), where they consistently show strong compute-performance trade-offs. The focus of this work is to introduce, a principled design of ENNs for reinforcement learning (RL) by formalizing the problem as a linear program. In the context of this work, we did not want to demonstrate scaling but show that the core principle works on standard RL benchmarks and the budget parameter allows interpretable control over the compute used, as can be seen in Appendix B.
>
> **2. Seeds and robustness**
>
> We agree with the reviewer that in some environments such as Hopper the confidence intervals are overlapping. However, our goal is not to show that we obtain large performance gains, but rather that **performance can be maintained while significantly reducing compute**. In this setting, overlapping confidence intervals are expected to be consistent with our claim.  The key result is that performance remains stable under reduced FLOPs, which is consistently observed across environments and algorithms. To follow best practices by [1], we can promise for a final submission to further evaluate the best hyperparameters with more seeds.
>
> **3. Compute allocation over long horizons**
>
> BEXA operates at the level of individual states. For each state $s$, it solves a constrained optimization problem to select an exit under the given budget. This means computation is allocated *locally per state*, in the sense that the **expected compute required per state is guaranteed to be under the budget constraint** assuming no function approximation error. As a result, the method is naturally agnostic to trajectory length and state distributions. If of interest, we can sketch a theoretical proof introducing a concentration bound of the compute used as the number of decisions (or the horizon length) increases during the discussion.
>
> **4. Interpretability of exit decisions**
>
> We agree that understanding which part of the state triggers early vs. late exits is an interesting analysis and important for interpretability. The behaviour is already in a sense interpretable as the budget allocation problem is described as a linear program. The obtained solution will set the probability of using an exit based on a trade-off between the estimated performance of an exit versus the computational resources required while comparing this across all exits. In our view, the complexity comes from the performance estimate aka the learned critic functions. We will consider adding a plot showing the distribution of Q-values and the obtained probabilities for some states. We note that prior work on ENNs has already shown that exit decisions correlate with input difficulty.
>
> **5. Hyperparameter sensitivity**
>
> Our method introduces two main hyperparameters:
> - a **budget parameter** (normalized in $[0,1]$), which is simple and intuitive to tune
> - a **loss weighting parameter** across exits, which is standard in early-exit architectures
> We include a sensitivity analysis of the budget parameter in Appendix B. The budget parameter does exactly what it should do. In practice, most of the variance comes from the underlying RL algorithms (SAC/TD3), not from BEXA itself. To ensure fairness, we used a fixed hyperparameter tuning budget across all methods.
>
> **Questions:**
>
> **Q1.**
> In principle, the gating mechanism could be formulated as an RL problem, where the internal policy selects exits to balance return and computation cost. Yet, this introduces challenges we will happily discuss during the rebuttal!
>
> **Q2.**
> This is problem-dependent and the network learns this behavior automatically. As described above, the budget constraint formulation will achieve a very interpretable trade-off between the estimated performance and the computational resources of an exit. Thus if we estimate that later exits do not result in an overall better performance, thus they have a higher Q-value, they will be not used. During initial testing we empirically observed that the agent used mostly the first and the last exit. As stated above we consider adding a plot during the rebuttal to highlight the practical properties.
>
> **Q3.**
> Allocating computation over full trajectories introduces several challenges, as stated above. Our per-state formulation avoids these issues by making local state based decisions.
>
> [1] Eimer, Theresa, Marius Lindauer, and Roberta Raileanu. "Hyperparameters in reinforcement learning and how to tune them." 2023.
>
> [2] Rahmath et al. "Early-Exit Deep Neural Network: A Comprehensive Survey." 2024.

---

> > ### Author Rebuttal · Reviewer_yrTV · 2026-03-31
> >
> > I thank the authors for answering my questions and for promising to add more runs (seeds) in their experiments. This partially resolves my concerns about empirical evaluation. I still think the empirical evaluation has a limited scope with only 5 environments from the same domain. It would have been great if we could also see the results transfer to other continuous control environments, such as the DM control suite domain. I increased my score to reflect the authors response.

---

### Decision · Program_Chairs · 2026-04-30

**Decision:**

Accept (regular)

**Comment:**

This paper explores the idea of early exit in neural networks for adaptive computation in DRL. While this idea has already been explored in CV and NLP, it is the first application in RL. There are no major concerns that are worth rejecting this paper. There were some concerns about wall-clock time reporting, but the authors have addressed them in the rebuttal. The number of seeds used in the experiment is less. I strongly encourage the authors to run more seeds for this final version (given that the number of tasks used in this paper is already low).